# Use of Machine Learning to Automate the Identification of Basketball Strategies Using Whole Team Player Tracking Data

**Changjia Tian [1], Varuna De Silva [1,*], Michael Caine [2] and Steve Swanson [2]**

[1] Institute for Digital Technologies, Loughborough University London, Loughborough University, London E15-2GZ, UK; tian.changjia@hotmail.com

[2] Institute for Sports Business, Loughborough University London, Loughborough University, London E15-2GZ, UK; Caine@lboro.ac.uk (M.C.); S.Swanson@lboro.ac.uk (S.S.)

**\*** Correspondence: varunax@gmail.com; Tel.: +44-742-6834139

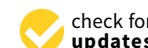

**Featured Application: For Sports Analysts, and similar, to deploy a novel machine learning process to determine strategies in basketball derived from player tracking data. Such analysis will assist coaching staff to prepare players and to formulate match strategies and team tactics.**

**Abstract:** The use of machine learning to identify and classify offensive and defensive strategies in team sports through spatio-temporal tracking data has received significant interest recently in the literature and the global sport industry. This paper focuses on data-driven defensive strategy learning in basketball. Most research to date on basketball strategy learning has focused on offensive effectiveness and is based on the interaction between the on-ball player and principle on-ball defender, thereby ignoring the contribution of the remaining players. Furthermore, most sports analytical systems that provide play-by-play data is heavily biased towards offensive metrics such as passes, dribbles, and shots. The aim of the current study was to use machine learning to classify the different defensive strategies basketball players adopt when deviating from their initial defensive action. An analytical model was developed to recognise the one-on-one (matched) relationships of the players, which is utilised to automatically identify any change of defensive strategy. A classification model is developed based on a player and ball tracking dataset from National Basketball Association (NBA) game play to classify the adopted defensive strategy against pick-and-roll play. The methodology described is the first to analyse the defensive strategy of all in-game players (both on-ball players and off-ball players). The cross-validation results indicate that the proposed technique for automatic defensive strategy identification can achieve up to 69% accuracy of classification. Machine learning techniques, such as the one adopted here, have the potential to enable a deeper understanding of player decision making and defensive game strategies in basketball and other sports, by leveraging the player and ball tracking data.

**Keywords:** sports analytics; basketball; data-driven strategy learning; multi-agent systems

## 1. Introduction

Strategy in high performance sport is increasingly data-driven, with analytical techniques transforming the way coaching decisions are made in the modern era. Machine learning algorithms are used to identify complex patterns in the data, which in turn allow meaningful classifications and predictions of future sporting events. Early adapters to this approach in team sports were those comprising isolated individual performances (e.g., baseball), however technological advances have enabled attacking sports with fluid collective movement (e.g., basketball, soccer) to make massive

strides in this area as well. For example, the National Basketball Association (NBA) now captures optical data from cameras positioned directly over the playing floor in every arena. is the most competitive basketball league in the world, comprising 82 games spanning across approximately 24 weeks [1].

Sports analytics involves data-driven modelling of sport, including managing physical performance of athletes, understanding game strategies and developing team tactics. Wearable sensors that measure players' movements, physical load and impact during collisions, combined with multi-view cameras that capture the entire field or court are routinely used to track players and ball movements in professional/elite team sports. The analysis of basketball data to gain competitive advantages is of interest to all the clubs, and is linked to the financial success of a team [2]. The quantitative analysis of sports especially basketball is a branch of science, which has grown initially through non-academic work [3], and has received extensive academic interest in the past decade.

A basketball game is modelled as a process by which the players that form the dyads attract to and repel from each other to produce the unique patterns that characterise player behaviours [4]. The team behaviour in basketball can be characterised by the space creation dynamics that relate to offensive behaviour [5] and the space protection dynamics that work to counteract space creation dynamics with defensive play [6]. Pick-and-Roll (PNR) has the highest frequency of occurrence of all space creation dynamics in basketball [5]. Understanding game strategies from past events (e.g., historical match analysis) may enable teams to gain a competitive advantage by knowing their opponents and by coming up with novel strategies to mitigate the perceived strengths of the opposing team. In the past, researchers have predominantly used two types of data to analyse basketball game strategies, i.e., play-by-play data that describe different events that happen on court such as shots, passes, dribbles, and fouls and player and ball tracking data. For example, play-by-play data can be used to learn the effectiveness of different types of PNR plays [7], or the factors that influence the effectiveness of inside-passes [8]. Wu and Swartz [9] proposed corrections for substitution errors using logistic classification models based on play-by-play data obtained from the National Basketball Association (NBA), in the USA. Talukder and Vincent addressed the problem of in-game injury prediction in basketball, where they combined play-by-play data, player workloads, and tracking data to build a Random Forest classification algorithm to predict in-game injuries for NBA players [10]. Likewise, Drazan et al. [11] investigated the performance of a basketball team. They used statistical techniques for analysis and plotted a heat map of the performance of players for the data visualization [11].

Tracking data, capturing player and ball movements are being widely utilised to inform these strategies. Analysis of tracking data is most useful in team sports where the spatial organization of the teams relative to the ball, and the temporal dynamics of the team organization, plays a significant part of the game strategy. As such there has been significant interest to learn game strategies from player and ball tracking data in team sports such as basketball and football (soccer). Past studies have shown that the trajectory data of in-game players can be considered as one of the most effective factors available in machine learning techniques: for example, mapping NBA strategies based on player-tracking data [12], analysing players' shooting style in the NBA using trajectory and body pose [13] and predicting three-point shooting results using recurrent neural networks [14].

Another important avenue of basketball analytics research is focused on predicting the outcome of a basketball play. Often such predictions are performed by developing generative probability models [15] that capture the contextual knowledge from performance analysis studies such as [4,16,17]. Furthermore, play-by-play data are an important element in developing useful generative models. However, performance assessment studies of basketball and the associated play-by-play data generated using player tracking data are heavily focused on offensive play metrics [18], and until recently modest effort has been spent on characterizing defensive play. This is mainly due to the ease of recording points, assists and related goal-scoring statistics. The availability of contextual information about defensive plays is thus an important need for development of useful analytical models.

The objective of this paper is to demonstrate the use of machine learning to recognise two defensive strategies commonly employed against the popular Pick-and-Roll (PNR) strategy in basketball. The paper proposes a methodology to recognise these defensive plays by analysing the spatio-temporal patterns in player and ball tracking data. The majority of tactical analysis studies in basketball investigate the status of a specific group of players rather than the whole team. The purpose of this study was to develop a computational model using machine learning algorithms to analyse in-game player movement during basketball match-play. Specifically, this paper utilises machine learning algorithms on player and ball tracking data in basketball to analyse the relative movement of in-game players and the ball to explain the strategies employed by the defensive team. The outputs of this work could be used to inform coaching decisions and to devise opponent-specific match strategies and team tactics. In the longer term, it is envisaged that similar techniques may be developed and deployed in a wide range of sporting and non-sporting contexts.

The contributions of this paper are as follows:

- Development of an analytical model to match the one-on-one relationship between players in both teams based on the players' location data. The process of matching is based on the underlying factors behind the trajectory data of players, such as distance, speed vector and the specified zone of each player.
- Development of a dynamic strategy classification model that automatically identifies change of defense mechanisms and classifies defensive tactics against PNR into two classes: 'switch' and 'trap' [19]. In comparison to previous work, the current study focuses on all players in the team (on-ball players and off-ball players).

The rest of this paper is organized as follows: Section 2 presents the work associated with data-driven algorithms for learning strategies in basketball. The proposed methodology for learning defensive strategies in basketball utilising player tracking data is presented in Section 3. Section 4 discusses the experimental setup and results and Section 5 comprises indications of future work.

## 2. Background on Basketball Strategies and Related Research

### 2.1. Team Strategies in Basketball

In this research, the 'pick-and-roll' and 'non-pick-and-roll' are focused upon. To differentiate between these two strategies, it is necessary to describe related basketball terms, such as one-on-one defense, zone defense, triangle-and-two, trap and switch, and help defense, as follows:

#### 2.1.1. One-on-One Defense and Zone Defense

One-on-one defense: Each defensive player defends one offensive player. As offensive players move, the defensive players who are assigned to specific offensive players move accordingly.

Zone defense: In contrast to one-on-one defensive strategies where defenders are assigned to specific opposing players, defenders in a zone defense focus first on defending specific areas (zones) of the floor [20]. When an offensive player moves into a defender's assigned area, the defensive player traditionally defends that player with one-on-one principles (until the offensive player vacates).

While 1-on-1 defense is popular in senior teams, zone defense is popular among youth teams such as Under-16 or Under-18 [21]. Combination or hybrid defense where a mix of man-on-man and zone defense is also utilised in some basketball settings. There are different types of combination defense systems such as Box and one, Diamond and one, and triangle and one. For example, in triangle and one method, two players are left to match up man-to-man, while the remaining three defenders protect against penetration by forming a triangle.

On the other hand, full court press is where a defensive team applies pressure on the offensive team the entire length of the court, through man-on-man or zone defense. Half court press allows the offense to arrive halfway down the court before applying defensive pressure.

### 2.1.2. Pick-and-Roll

The general pick-and-roll, also known as on-ball screen [20], involves one offensive player setting a screen (pick) for a teammate in possession of the ball (on-ball player). As shown in Figure 1, it contains three steps: Firstly, the off-ball player sets a screen for an on-ball player; Secondly, the on-ball player reads defensive strategies and uses the screen to create an open-shot opportunity for their teammates or themselves; Thirdly, after screening, the off-ball player reads the defense and creates an open-shot opportunity for their teammates or themselves.

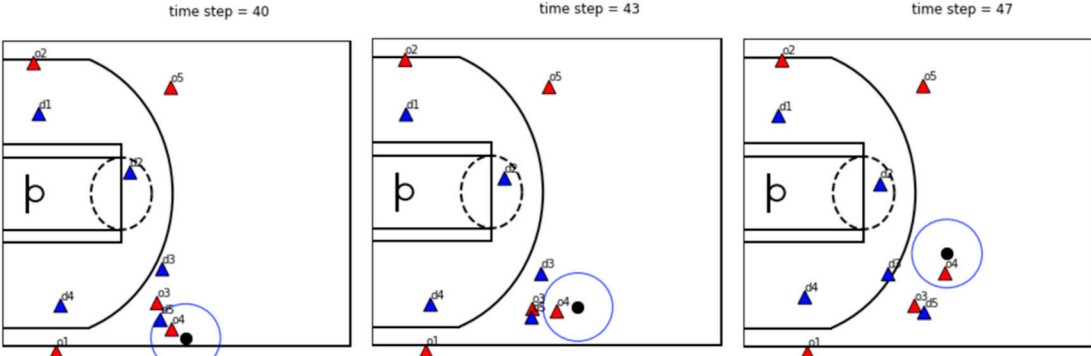

**Figure 1.** In these 3 timesteps, o4 and o3 play a pick-and-roll. Firstly, off-ball player(o3) sets a screen for on-ball player(o4). Secondly, o3 uses the screen and d5, who is the original defender of o3, is blocked. Thirdly, via screening, o3 creates an open-shot opportunity for o4, and is now defended by d5.

The pick-and-pop is a variation of the pick-and-roll strategy, where the player does not roll to the basket, instead popping out to the perimeter. On perimeter the player gets an open look at the basket to shoot, when the pass is received from the guard.

### 2.1.3. Switch and Trap

For defensive players, two options named 'switch' and 'trap' are often deployed to defend pick-and-roll. According to McIntyre [19], 'switch' denotes an on-ball defender and off-ball defender(s) switching their original matchups as shown in Figure 2 [22], while 'trap' means both on-ball defender and off-ball defender(s) double the on-ball player (i.e., two or, very occasionally, more, players directly defending the player with the ball) as shown in Figure 3.

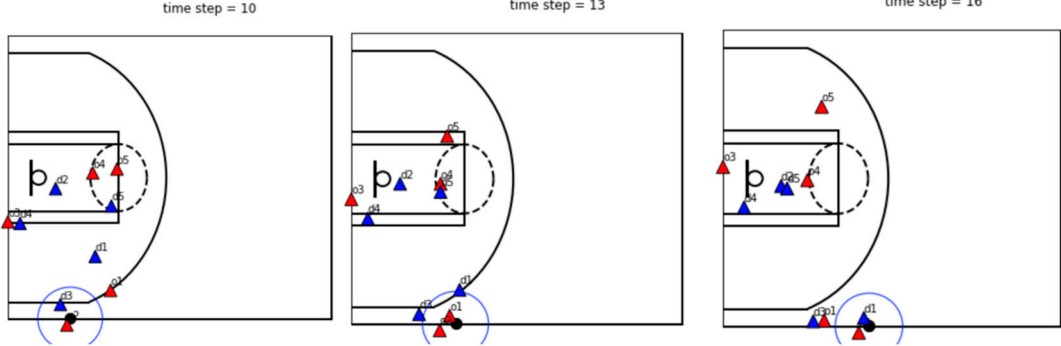

**Figure 2.** In these three timesteps, o2 and o1 play pick-and-roll. After that, d1, which is the original defender of o1, becomes the defender of o2, and d3 becomes the defender of o1. This process is called 'switch'.

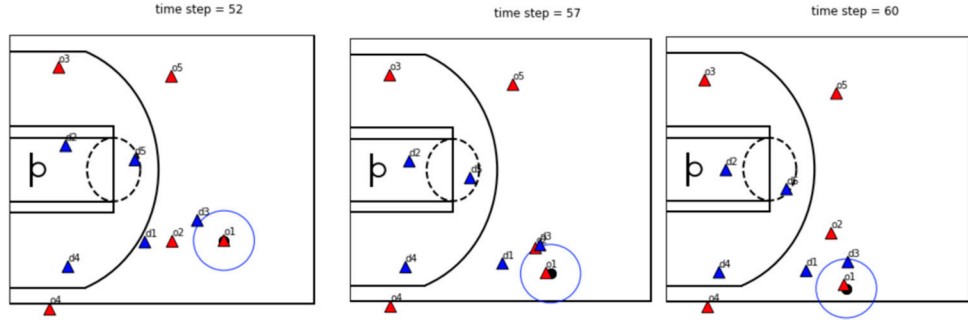

**Figure 3.** In this case, o2 and o1 play pick-and-roll. After that, both d1 and d3 try to defend o1 (on-ball player), and at the same time, o2 (off-ball player) make space for themselves. This process is called 'trap'.

## 2.2. Data Driven Performance Analysis in Basketball

Analysis of Performance in basketball is a complex process that involves substantial number of dynamical interactions between technical, tactical, fitness, and anthropometric characteristics of players [23]. Analysis of the intra- and inter-couplings of the playing dyads (i.e., coupling between two players) in basketball, obtained through player tracking data retrieved from video footage, revealed good evidence for dynamical relations in both the longitudinal (basket-to-basket) and lateral (side-to-side) directions. A basketball game can be seen as the process by which the players that form the dyads attract to and/or repel from each other to produce the unique patterns that characterise their own behaviours [4]. In [16] authors report an analysis of game behaviour in basketball by examining the interaction between two teams, by analysing the phase difference of two temporal features of the spatial arrangement of the teams, i.e., spatial centres of the teams and stretch index. As expected, relative phase analysis of the spatial centres indicated a strong in-phase relation between the two teams in both the longitudinal and lateral directions. In comparison, the stretch index which measure the mean deviation of individual players from the spatial centre, indicated in-phase relationship only in the longitudinal direction.

Lamas et al. [5] presented the concept of space creation dynamics, which is set of basketball offensive classes, corresponding to the possible offensive tactics to create space in the adversarial defensive system that leads to scoring opportunities. The corollary of space creation dynamics within the scope of defensive tactics is the definition of space protection dynamics to counteract the SCDs [6]. The interactions between space protection dynamics (pertaining to defensive actions) and space creation dynamics (pertaining to offensive actions) was analysed in [6] as a method for tactical evaluation of team playing patterns in basketball.

Ball screens are an important offensive tool used by teams in basketball, where a screener sets the screen generating an advantageous situation to the dribbler who will pass it on to a teammate who is in an open field-goal situation or shoot without the defensive pressure. This action where two players work in tandem is commonly referred to as Pick-and-roll [7]. In [24], authors investigated the predictors of success of ball screens such as time, space, players, and tasks performed, and illustrated the relationships of ball screens and offensive success with the orientation of the ball screen and actions of the dribbler after the screen. By considering a sample of 668 ball screens from 17 games of the Spanish basketball league, the contextual factors that affect the ball screens such as, score-line, offensive system, duration and game quarter, were analysed in [25]. Through statistical analysis of play-by-play data related to the ball screens, it was concluded that the effectiveness of the ball screens was greater when there was high time pressure, especially towards the final few seconds of a ball possession.

Pick-and-Roll (PNR) has the highest frequency of occurrence of all space creation dynamics in basketball [5]. A total of 12,376 pick-and-rolls in European basketball league were analysed in [7], by considering the play-by-play data from Synergy sports systems. Different types of PNR plays are analysed for their effectiveness and the authors conclude that possessions that end with the screener's

rolling in the shot and those that end with 2 passes following the PNR are the most effective uses of the PNR and the least successful type is when the ball handler shoots [7].

The collective behaviour of players for the inside pass events was analysed in [8], where it was shown that the interactions combining passer's previous actions (dribbling or faking) with receiver's cuts towards the basket achieved the highest offensive effectiveness. Performing screens in favour to the receiver was an effective alternative to increase inside passing options since it reduces the defensive [8]. The effect of space among players towards team performance is analysed in [26].

A review of different contextual factors that affect a basketball game is presented in [17]. One of the key contextual factors discussed in [17] is the period of the game, where literature consistently suggest that offensive effectiveness decreases throughout the game due to the increase of defensive pressure [27]. A greater offensive effectiveness was seen at the beginning of a game where teams adapt a faster game pace with shorter possession durations and less passes. On the other hand, playing longer possessions and involving more players increased the scoring chances towards the later parts of the game.

Most of the scientific literature on basketball analysis has mainly focused match events without integrating information regarding the observed behavioural patterns and team strategies [28]. Another important avenue of research is on predicting the outcome of a basketball plays. Often such predictions are performed by developing generative probability models [15] that capture the contextual knowledge from performance analysis studies such as [4,16,17]. For example, authors in [29] used a possession-based Markov model to model the progression of a basketball match, where the model parameters were estimated from NBA play-by-play data and from the teams' summary statistics. The team interactions and strategies are modelled as a dynamical system to represent different match situations in [28]. A similar model that can be used for basketball match simulation is found in [30]. The model in [30] is based on a graphical model that encodes players on the court, their actions, events as nodes on a graph and the edges of the graph denote the possible moves in the game flow. The model parameters in [30] are estimated using player tracking data, play-by-play data and team lineup data from 2013–2014 season. However, due to the complex nature of the basketball strategies, developing a model of team dynamics is an extremely complex problem, and inclusion of many contextual factors is one way to improve the accuracy of such systems.

### 2.3. Use of Machine Learning in Basketball Analytics

Wang and Zemel [31] developed a machine learning model to process player tracking data to identify offensive plays in basketball. The trajectory dataset utilised in [31] was provided by SportVU [32]. Each data point within this dataset is stored as a sequence which has the coordinates of the ball and all players on the basketball court. A sequence is recorded every 0.16 s. The authors devised variants of neural network algorithms to model a classifier to recognise basketball tactics based on unlabelled historical data. This study implemented the recurrent neural network (RNN) which can deal with sequential data of variable length. The inputs of the RNN comprise the player coordinates and the outputs are the labels for the different offensive tactics. The authors also describe the development of an autoencoder neural network based on an extra dataset provided by NBA team, the Toronto Raptors, to enhance the model to become a player-specific model. The researchers demonstrated that the model could be used to predict the movement of players, i.e., to forecast future player position [31]. This recurrent neural network achieved a classification performance of 80.6% accuracy. Nevertheless, due to the limited number of training data, the predictive capability of the recurrent network was relatively weak (less than 60%).

Mclntyre et al. [19] also developed a supervised machine learning algorithm utilising NBA player tracking data from SportVU. The authors focussed on recognising and analysing ball screen defence (pick-and-roll). Their system takes the unlabelled trajectory data as input, and then derives the time of all ball screens that occur during each game. Finally, classifying how the ball screen was guarded by defence players. In [19], the researchers first defined four players who formed the ball screen tactics: a

ball handler, a ball defender, a screener and a screener defender. Secondly, they defined four different categories for the defensive tactics based on the trajectories of these four players. In [19] authors trained a logistic regression machine learning model that uses the distance between four defined characters as features. Finally, through 5-fold cross-validation they achieved an overall average classification accuracy of 69%. In addition, this paper analysed the trends of different players and different teams in the face of the ball screen.

Learning long-term behaviour for multi-agent spatiotemporal trajectories is a key challenge in many learning problems [33]. Zhan et al., proposed Multi-Agent Generative Behavioural Cloning, referred to as MAGnet, based on a deep learning algorithm. The authors describe MAGnet as a flexible class of generative models that can generate rich multi-agent spatiotemporal trajectories over a long-time horizon [33]. Moreover, the main advantage of MAGnet is that it has a shared structure between agents, and it has a hierarchical latent structure to jointly represent long-term (macro) and short-term (micro) temporal dependencies. Furthermore, the researchers compared MAGnet with a Variational Recurrent Neural Network algorithm (VRNN) [34]. The VRNN algorithm proposed by Chung et al. [34] is built by the combination of a recurrent neural network with variational autoencoder. According to [33], the MAGnet algorithm has significantly higher performance than the VRNN model [34]. The concept of macro-goals allows the observer to analyse the long-term goals of a player and how they change their objectives during game play. Zhan et al. suggested that exploring a more powerful probabilistic structure to handle more agents under a complex condition is an avenue worth pursuing.

Nearly all basketball tactical analysis studies only analyse the status of a section of players and not the whole team. For example, the pick and roll strategy typically comprises four players—two players on each team—the remaining six players are not analysed. Another example is where the study of shooting outcome predictions only takes into account the tracking data of one player. However, basketball is a team game consisting of ten players, therefore, if only the status of some players is analysed, the results of the analysis will be compromised by the inevitable interference from undefined factors.

## 3. Methodology

This section describes the proposed methodology for dynamic (automated) recognition of defensive strategies in basketball. This section is organized in three subsections: the dataset, the analytical model for defensive relationships between players and the data-driven classification model to recognise defensive strategies based on the identified defensive relationships.

### 3.1. Dataset

SportVU tracking data, provided by STATS LLC, was utilised [35]. The dataset contains basketball player trajectory data obtained from the NBA. Each data file represents a possession. Possession is a sequence of game events during which one team retains the ball and ends when the ball is captured by the opposing team, or if a shot at the basket is made [33]. Possession is an essential efficiency statistic because it allows statistical analysis based on a per-possession basis. The dataset obtained from [36] consists of possessions of variable length: each possession is a sequence of tracking coordinates ($s^i_t = x^i_t, y^i_t$) for each player *i*, recorded at 25 Hz, where one team has continuous possession of the ball. The possessions last between 50 and 300 frames. The dataset obtained contains 36,330 possessions are taken from the STATS SportsVU tracking data from the 2012–2013 NBA season. These data have come from approximately 630 games. It should be noted that in the original study that used this data [36], 80,000 possessions were analysed along with the event data, of which only the 36,330 possessions are made publicly available, and excludes the associated event data. Of the 36,330 possessions, only 32,377 possessions were used in our study.

All the data frames were standardized in-to attack-defense situations, regardless of the side of the court the activity is happening. Therefore, all activity can be analysed as taking place from one

half of the court. The proposed method to identify defensive zones, utilises a $400 \times 360$ row-major grid to represent a basketball half court. For each time step, the player and ball positions are stored as serial indices in a $400 \times 360$ grid representation of the half-court, indexed row-major, with $(0, 0)$ in the top-left [33]. The dataset utilised in this study, is provided only for half-court, and hence cannot be used for analysing defensive strategies such as full court presses.

The objective of the paper was to use machine learning to automatically classify switch and trap by analysing the player and ball tracking data, and for this the labelled data need to be provided. An analyst should ideally go through the individual frames of a possession in sequence to recognise if there was a switch or trap. However, this was a very time-consuming task to label all the possessions. Therefore, we developed an analytical model that is explained in Section 3.2 to analytically identify and label parts of possessions that contain a switch or a trap. The analytical model selects 5 frames at a time from a possession and analyse if there is a switch or trap in the possession. Not all the possessions in the dataset will contain a switch or a trap, and notice that some possessions will contain more than one occurrence of switch/trap. Finally, a total of 42,865 plays of switches and traps were identified from 32,377 possessions.

144 possessions of this data set that contained a switch or trap strategy was manually selected and annotated with an appropriate label by one researcher. These manually labelled possession data are used for identifying the reliability of the analytical model.

### 3.2. Analytical Model for Labelling the Dataset

Objective of this model is to develop a labelled set of switch and trap plays identified from the tracking data to be used as a training set. For this purpose, the analytical model encodes contextual knowledge about basketball defense as described in Section 2.

A basketball game consists of two teams each with five players. The team that possesses the ball at any given time is considered the offensive team, and the team without the ball is considered the defensive team. Typically, each defensive player selects one matched offensive player to defend. An experienced player, coach or spectator can identify the correspondence between players (who is defending whom) based on the position of players. The objective of this algorithm is to understand the defensive relationship between the players of the two teams based on the player tracking data. This model underpins the overarching aim of this study by enabling the classification of defensive strategies in basketball.

The analysis of the underlying relationship between in-game players needs to be a team-level analysis rather than a specific player-level analysis. Determining the one-on-one defensive relationship between players will act as the foundation of the proceeding classification model building. The analytical model is based on three different attributes: the location data at each time step, the distance between players, and the defensive zone of players. The structure of the proposed analytical model is illustrated in Figure 4.

Firstly, in terms of the location at each timestep, a data example means a possession and a possession consists of many consecutive time-steps. In addition, the interval between each time step is 0.16 s (6.25 Hz). Through processing the raw data, we obtain the location of all the ten players at a given timestep. Utilising these location data it is possible to calculate the velocity of each player. The velocity is utilised to identify the similarity of movement of two players at consecutive time-steps.

Secondly, the model utilises the distance between players. The offensive player who is closest to a particular defensive player has a high probability of being the defensive target (match) of the defender. A good defense requires the defensive player to place a lot of pressure on the opponent, therefore in this process, defensive players usually approach the defensive target to block the movements. Additionally, in some situations, if an offensive player has a low offence threat, the defending opponent of this offensive player may keep a farther distance. For example, if an offensive player is far from the three-point line, and the long-range shooting ability is poor, the defender of this offensive player may keep a long distance between them. In this scenario, while the defensive player would be distant from

the offensive player, they would still, most likely, position themselves between the offensive player and the basket. Once the offensive player approaches the basket, the attack success rate will increase, which also means a greater threat to the defensive team. Therefore, this system adds a feature that identifies the offence route of the offensive player to further inform the defensive relationship. When there is no defender around an offensive player, the system will connect the straight line between the offensive player and the basket because the straight path of the player to the basket is the shortest and most threatening path.

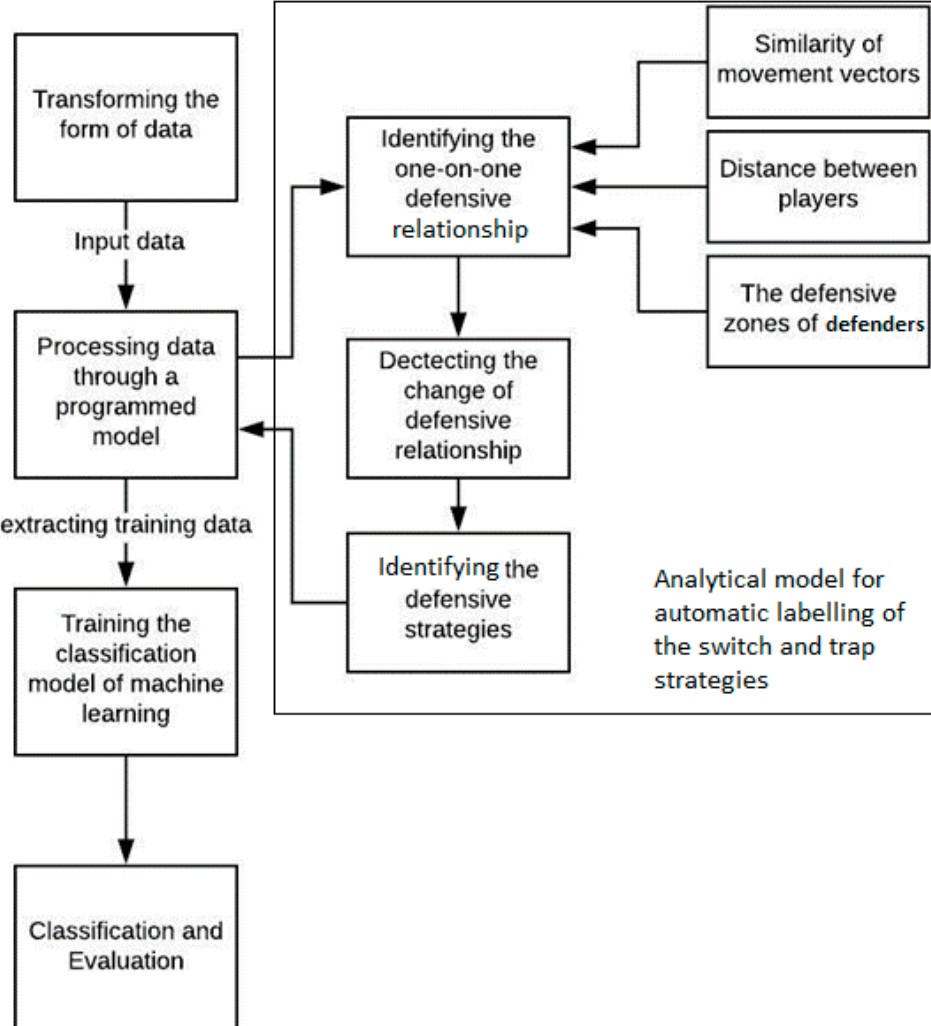

**Figure 4.** Summary of the proposed workflow to use machine learning to recognise defensive plays against pick-and-roll from player tracking data.

Finally, the analytical model utilises the notion of a 'defensive zone' of players, which separates the basketball court into five zones. The area of each defensive zone is evenly distributed according to the distance between players. The method to identify defensive zones, will calculate the distance from each point on the grid to the five defensive players when constructing the defensive zone of the defensive players. Next, the nearest defensive player is assigned to a given point (x, y location) on court. The Figure 5 illustrates the defensive zones identified by this algorithm for a given moment when defensive players located at black points. Different colours indicate the defensive zone assigned to each player. A defensive player is considered to be 'in charge of' one zone. If one or more offensive players enter zone of a certain defender, then the algorithm will re-analyse to determine if the one-on-one defensive relationships should be changed.

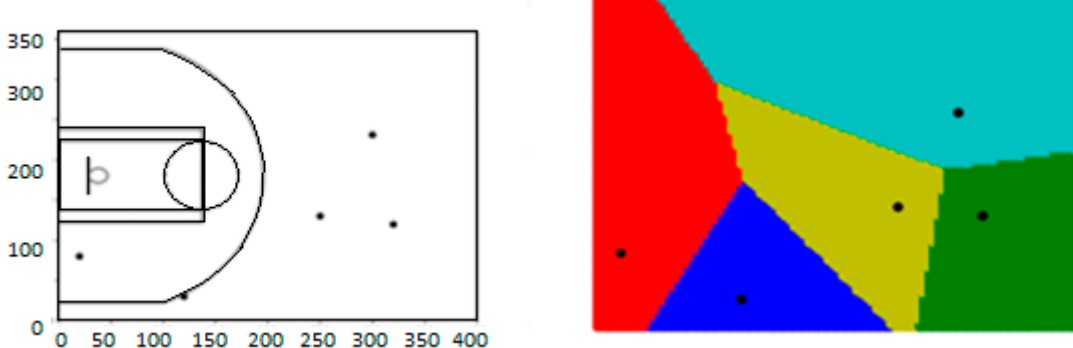

**Figure 5.** The defensive zone of defenders with different sample interval of 2 pixels.

According to the above three features, the analytical model in Figure 4, matches each defensive player with an offensive player. The model will observe the defensive relationship throughout a specific possession to identify any defensive relationship changes. The next subsection describes the data-driven model that identifies and classifies different defensive strategies based on the one-on-one player relationships.

### 3.3. Data Reliability Procedures

In the experiment, a single researcher selected and annotated 144 sample data trajectories. The analytical model analysed 32,377 trajectories. The reliability of the analytical model in recognising switch and trap was assessed, against the 144 sample data trajectories that were manually labelled with switch and trap. As such, the reliability measurement is similar to the way inter-observer consistency measurement is performed. The consistency is analysed for the subset of 144 random possessions that were manually annotated. Reliability ratios are evaluated according to the percent agreement value which was >0.7 for all categories [37]. It was deemed that Cohen's Kappa statistic was not appropriate in this instance as one observer is a deterministic computer model, which does not show uncertain annotation behaviour expected in human annotators.

### 3.4. Classification of Defensive Strategies

The Section 3.2 described the methodology to identify and label the switch and trap defensive relationships and any changes during a given possession. Every time a defensive relationship change is detected, the strategy utilised within the play segment needs to be identified. This section presents the classification algorithm to recognise the defensive strategy utilised.

We evaluated two different sets of training features for building this classification algorithm. The first approach was based on the location of all players, given as (x, y) coordinates of individual players. The second approach is focused more on the players whose defensive relationship change during the possession, with limited focus on the rest of the players. This is motivated by the fact that change of relationships might flag the start of a new strategy. In this second approach, we consider eleven features that contain the movement vectors of the four players whose defensive relationship changes (two from defense and two from offense sides), the distance from the centre point of the four players to the basket, and the distance from the other six players to that centre point. The feature vectors used in the above two approaches are illustrated in Figure 6, where F'X' on the first row illustrates the Xth feature.

| F1 | F2 | F3 | F4 | F5 | F6 | F7 | F8 | F9 | F10 | F11 |
|----|----|----|----|----|----|----|----|----|-----|-----|
| | Offence Team | | | | | Defense Team | | | | |
| Ball Position (x, y) | Player 1 Position (x, y) | Player 2 Position (x, y) | Player 3 Position (x, y) | Player 4 Position (x, y) | Player 5 Position (x, y) | Player 1 Position (x, y) | Player 2 Position (x, y) | Player 3 Position (x, y) | Player 4 Position (x, y) | Player 5 Position (x, y) |

(a) Approach 1

| F1 | F2 | F3 | F4 | F5 | F6 | F7 | F8 | F9 | F10 | F11 |
|----|----|----|----|----|----|----|----|----|-----|-----|
| Movement vectors of players whose defensive relationship changed | | | | Distance from mid-point of O1, O2, D1, D2 to basket | Distance of individual players to the mid-point of O1, O2, D1 and D2 | | | | | |
| | | | | | Offence Team | | | Defense Team | | |
| Offence Player 1 (O1) | Offence Player 2 (O2) | Defence Player 1 (D1) | Defence Player 2 (D2) | | Distance from O3 | Distance from O4 | Distance from O5 | Distance from D3 | Distance from D4 | Distance from D5 |

(b) Approach 2

**Figure 6.** Different feature vectors used for classification machine learning algorithms.

We investigated several machine learning models as candidates for classification. These include K-Nearest Neighbour algorithm (KNN), Decision Trees and Support Vector Machine (SVM), which provide varying levels of complexity and classification capabilities.

## 4. Experimental Results and Discussion

### 4.1. Selection of Classification Model

We used 10-fold cross-validation to train and validate the classification model. The 10-fold cross validation method separates the whole dataset into ten sub-sets. Selecting one of the ten sub-sets iteratively as a validation set, and other nine as the training sets. The average accuracy of the 10 rounds, for each algorithm is used as a guide to select between different models.

Two feature sets were investigated for training the classification model. In the first approach considered only the location data, and the second approach used both movement vectors and the distance to be basket. Several machine learning models were utilised to compare the two approaches for classification as described in previous section. The results are summarized in Table 1.

**Table 1.** Accuracy of Different Classifiers on two approaches of defensive strategy classification.

| Algorithm | Approach 1: Location Data | Approach 2: Movement Vectors and Distance |
|-----------|---------------------------|-------------------------------------------|
| *Quadratic SVM* | 49.7% | 68.9% |
| *K-NN Classifier* | 51.8% | 71.5% |
| *Decision Trees* | 46.2% | 65.6% |

As illustrated in Table 1, the second approach seems to be more suitable for most machine learning algorithms. The features of the first approach are not informative enough for the classifiers to learn the difference between the two classes.

After the 10-fold cross validation, the confusion matrix and receiver operating characteristic curve (ROC) of each algorithm can be computed. This study uses ROC to analyse the validation result. Area Under the Curve (AUC) means the probability that a randomly chosen positive example is ranked higher than a randomly chosen negative example, that represents the performance of this classification experiment. The model of the higher AUC value has the stronger classification capability. According to the comparison of the following three pictures, the KNN algorithm has the largest AUC value with

0.8. The classification capability of SVM is 0.02 stronger than Decision Tree. The confusion matrices and the ROC curves for the different classification algorithms are illustrated in Figure 7.

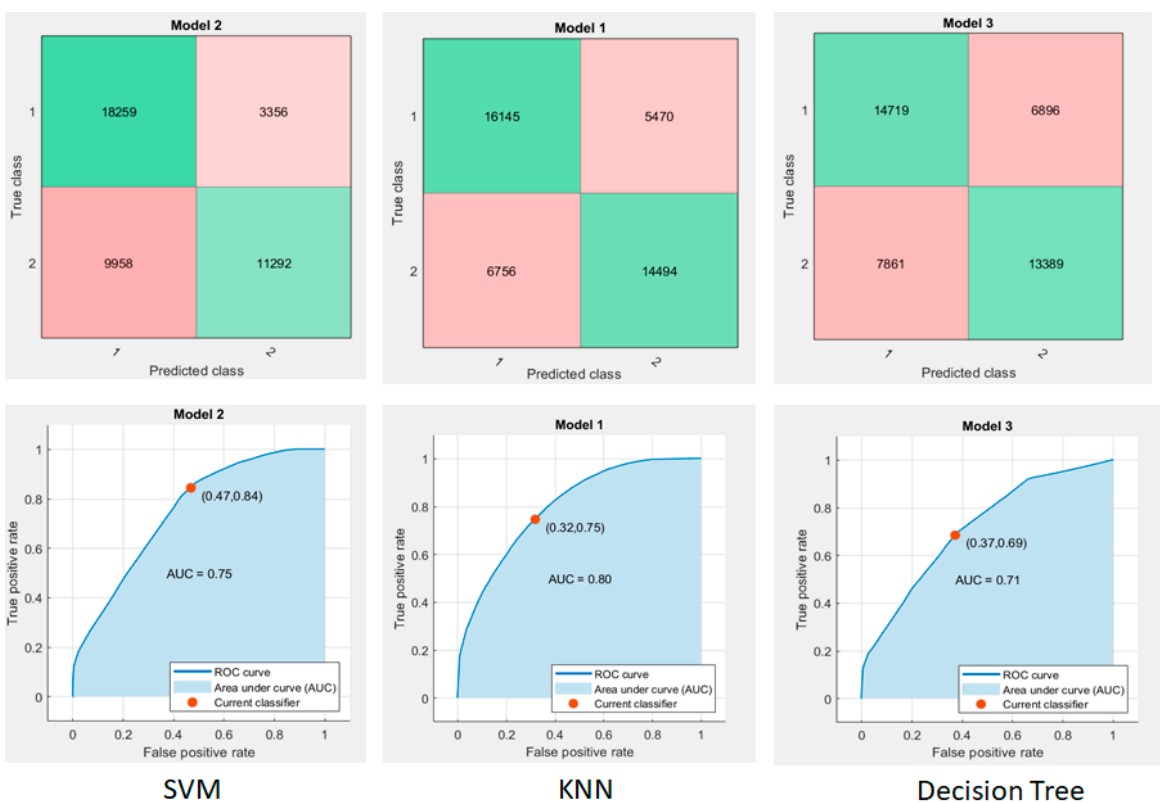

**Figure 7.** The receiver operating characteristic (ROC) curves and confusion matrices for different classification algorithms compared for defensive strategy classification.

### 4.2. Performance of the Machine Learning Model

The results of SVM is worse than KNN. While the KNN is suitable for low dimensional problems. The advantage of SVM is that it can handle high-dimensional features using linear functions [38] but requires a large dataset. Therefore, additional features were experimented with, such as adding the distance from ball to basket and the movement vector of players whose defensive relationship has not changed. The highest accuracy of the SVM algorithm has increased by 2% after increasing the number of features. However, as the number of additional features increases, the accuracy of SVM and other algorithms is gradually decreasing. Thus, the algorithm is experiencing curse of dimensionality. Furthermore, there may be several potential factors affecting the training model: the features are less effective, and the amount of data is insufficient. Therefore, we can optimise the training model by two methods: optimising the quality of features or expanding the amount of data.

In the present experiments, the KNN shows the highest accuracy levels. However, it should be noted KNN algorithm works by comparing a new data point against all the past data points, and assigning it for the majority voted class within the k-number of nearest neighbours. Therefore, for the KNN algorithm to be used, all the data points (e.g., 32,377) need to be stored on the computer. This is a significant disadvantage, especially when the dataset is extremely large. Therefore, considering both the computational complexity and the classifier performance, the SVM with the movement vectors and distance features, would be the best method to classify defensive strategies.

This algorithm can be improved in multiple ways. A player-specific dataset that contains information about players' abilities can be combined with the current trajectory dataset, which will strengthen the system's ability to define each defensive person and determine the defensive relationship between players. For example, the algorithm of this paper determines the defensive zone of players

based on the distance relationship. However, in reality, different players have different defensive capabilities. So, a player-specific model that takes into account the defensive ability of players or quantifies the defensive capability, will enable segmentation of the defensive zones according to the individual defensive capabilities of the players.

Another avenue of work would be to manually identify the one-on-one defensive relationships data, to develop a labelled dataset which can be used to classify the defensive strategy. Furthermore, to optimise the features and using a Recurrent Neural Network to classify temporal activity would be an intuitive way forward.

### 4.3. Application of the Proposed Method for Basketball Analysis

Team sport strategy enables individual and group actions to be organized in order to produce collective, creative and unpredictable execution of the team's actions [28]. Development of analytical tools that can be used to predict the outcomes of different game strategies is a trending field of study in sports analysis. In the past, extracting the player and ball location was a cumbersome process that involved extensive video analysis. Availability of player and ball tracking data is enabling the researchers to develop such analytical tools.

Due to the ease of recording points, assists, and related goal-scoring statistics, most play-by-play datasets often focus on offensive analytics [18]. Therefore, the presented model in this paper focused on the defensive tactics. Specifically, we focus on the defensive actions against pick-and-roll (PNR), which is the most commonly used offensive strategy in basketball [5]. The proposed machine learning model can identify two defensive tactics against PNR, known as switch and trap, by considering features derived from player and ball tracking data. The model achieved an overall accuracy of 69%. Although, the current model was trained to identify only switch and trap, the methodology can be replicated to learn other tactics, provided labelled training data can be sourced.

The proposed model can be used for different applications of basketball analytics. For example, it can be used for match analysis to automatically retrieve a given defensive strategy from player and ball tracking data. After retrieval, the analysts can see how different opponents defend PNR, and analyse the relative effectiveness of different defensive strategies. Such information can be very useful to design strategies before games to thus providing competitive advantage through analytics. Data driven ghosting schemes are quite a useful and emerging tool for analysing the defensive behaviour of a team [39]. However, as of present such schemes are strictly based on tracking information and some basic contexts such as team roles, but lack crucial important contextual information such as defensive types and game period. However, contextual information regarding different defensive moves cannot directly be accessed from play-by-play data available. Hence, our proposal can be used as a method to embed additional contextual knowledge for player tracking data.

There has been a significant interest in the sports analytics research community generative probability models for predicting the outcomes of games. These models are predominantly based on the available play-by-play data [29] and encode the knowledge of analytical studies on basketball performance [30]. However, the models such as [29], or [30], do not take in to account the complex nature of dynamical interactions between players. The play-by-play data that are collected for basketball matches do not encode many offensive or defensive strategies, but only the high-level events such as shots, passes, dribbles, chances and fouls. The presented tool to recognise defensive moves against pick and roll from player-tracking data is an example of a model that represents dynamics of a subset of players, can enrich game outcome prediction models that represent the match situations as a continuous dynamical system [28] or a graphical model [30].

In summary, the presented machine learning methodology is an important step toward the provision of additional contextual information that cannot easily be accessed through play-by-play data. The presented model to identify defensive strategies can augment such play-by-play data, especially if coupled with further analysis on the outcomes of the plays to provide additional metrics about defensive effectiveness.

### 4.4. Limitations of the Current Methodology

There are many offensive plays in basketball other than Pick-and-roll, and many defensive strategies against PNR other than switch and trap. The objective of this study is to the illustrate the possibility to identify a specific defensive strategy in a highly dynamical team setting, where the team roles may change over the course of a given possession, by utilising the player and ball tracking data. This model is not an exhaustive representation of all possible offensive or defensive strategies, but an important contribution especially towards the community of basketball analytics researchers who have taken an active role to develop models of game progression dynamics.

Basketball player and ball tracking data are not available for free, but play by play data can be obtained from NBA.com. Our study was based on a limited player-ball tracking dataset (36,330 basketball possessions), which is freely available from STATS LLC, without any corresponding contextual information such as play-by-play data associated with the tracking data. As an academic endeavour, we illustrate the possibilities that are present if data is made available freely. Availability of more data, along with the contextual information will expedite the progress of this field and widen the interest of academic researchers in the field.

Furthermore, the number of manually annotated data points (200 out of 32,377 total data points) considered within this study are also limited. It should be noted that the data annotation is an extremely time-consuming task, which requires the analyst to go through the tracking data multiple times. This is worsened when corresponding video footage is not available nor when the contextual information is missing. Machine learning algorithms generally requires large amounts of training data. In this study, we used an analytical model to automatically label the data, hence our labelled dataset is limited by the capabilities of the analytical model. Ideally, the dataset has to be labelled by an analyst, and availability of such labelled datasets is crucial for development of machine learning algorithms.

## 5. Conclusions

The objective of this paper was to classify defensive strategies in basketball by utilising spatio-temporal pattern recognition using player and ball tracking data. A tracking dataset of player and ball trajectories in 32,377 possessions from nearly 630 basketball games in 2012/13 NBA season from STATS LLC was used in the study. The objective of this study was to classify two common defensive strategies (known as switch and trap) used against a popular offensive strategy known as pick-and-roll, by considering different features from the play-and-ball tracking data. To develop a substantial annotated dataset, an analytical model was developed with a capability to automatically identify and label trajectories that contain switch and trap defensive plays. This analytical model that recognised defensive plays was based on three features: the similarity of the movement vectors of the players; the distance between players, and the defensive zone of the players. Subsequently, a classification model was learnt to classify the defensive strategies in situations where the defensive relationship of players changed. The model extracted raw location data and features derived from the location data such as (velocity/distance) to train the classification model, which was able to classify "switch" and "trap" strategies in basketball. Different types of classifiers were evaluated such as, Support Vector Machines (SVM), Decision Trees and K-Nearest Neighbour (KNN). Considering both classifier performance and complexity, SVMs were deemed the best solution, which produced a classification accuracy of 68.9%. While the current work considered only "switch" and "trap", there are many more defensive strategies involved with Basketball. Future research may also wish to consider alternative strategies for defending strategies such as pick-and-roll, and perhaps consider defensive moves against other offensive strategies such as pick-and-pop. Furthermore, alternative methods to label large spatio-temporal datasets would also lead to better outcomes, as compared to analytical method proposed in this paper.

The outputs of this study illustrate the suitability of player tracking data to learn competitive game-related strategies employed in team sports such as basketball. Data-driven methods such as the machine learning methodology presented in this paper, can provide useful insights into game play. Furthermore, such models may enable the improvement of analytical software by providing additional contextual information related to defensive plays, in addition to the play-by-play data that are commonly available. Thus, facilitating advanced analytical solutions that has the potential to inform player development, coaching strategies and game specific tactics.

**Author Contributions:** Conceptualization, C.T. and V.D.S.; methodology, C.T. and V.D.S.; software, C.T.; validation, C.T., S.S., and V.D.S.; data curation, V.D.S.; writing—original draft preparation, C.T., V.D.S., M.C., S.S.; writing—review and editing, M.C., S.S., V.D.S.; funding acquisition, V.D.S., S.S. All authors have read and agreed to the published version of the manuscript.

**Funding:** This research was funded by Engineering and Physical Sciences Research Council, grant number EP/T000783/1.

**Conflicts of Interest:** The authors declare no conflicts of interest.

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
