# Peer review of "Use of Machine Learning to Automate the Identification of Basketball Strategies Using Whole Team Player Tracking Data"

_applsci, doi:10.3390/app10010024_

Round 1

Reviewer 1 Report

Please see the attached document with my comments and concerns. This article needs to do a better job communicating how its results could be useful to coaches, players, and other basketball decision makers. It also needs to further elaborate on its data and methods.

Author Response

Dear Reviewer,

Due to your comments, we have made several significant changes, in to the resubmission of the manuscript. We would like to thank your for your considerate and thoughtful comments which you have provided voluntarily. 

We have provided additional information necessary for understanding our method, and here we would like to point the key changes in the current manuscript:

We have included a new section (subsection 2.2) on the work carried out in basketball analytics in general, as it was important to position our contribution. A total of 24 new references are included in the new manuscript. The introduction is improved to provide more background about basketball analytics, previous work, its importance and how our work fit in.  More background on defensive mechanisms is provided in section 2.1. More information about the data set is provided in section 3.1, and the data reliability measures are included in the new subsection 3.3.  A new subsection (section 4.3) under discussions is written about the usage of this model and the contribution in general towards the basketball analytics area.  A new subsection 4.4 is included to write about the limitations of the current study.  The abstract and conclusion is adjusted to reflect the above changes and to narrow down the focus of defensive strategies in general to defensive strategies employed against pick and roll. 

We have left all the new additions with track changes to highlight the changes. 

Below you can find a point by point response to individual questions/comments posed by you. 

Thanks again,

Authors

Review:

Review of: Use of Machine Learning to Automate the Identification of Basketball Strategies using Whole Team Player Tracking Data

Primary Suggestions:

I think it is necessary to provide a few sentences of background information on the NBA’s relevance and why defensive, analytical advantages in the NBA would be important (financially or otherwise).

Thanks for the suggestion, we have improved the introduction in the current draft and very specifically addressed this comment in lines 43-50, and 55-59. 

On page 3, section 2.1.1, two types of defense are mentioned—man-to-man and zone. However, there are hybrid defensive schemes like the triangle that go unmentioned.

I think this issue was mainly raised because in our original draft we were not very explicit as to what defensive strategies we are addressing. This has been rectified in the current draft, and we have discussed this in section 2.1.1, and the next text can be found from lines 151 to 159. 

Are full-court presses, as a defensive strategy, able to be analyzed? Given that the data is shaped to only examine the attacking half of the court, I would assume not, but this needs to be mentioned as a limitation of this study.

No this cannot be done, and now acknowledged as a limitation in lines 366 to 367. However, this is a limitation of the publicly available dataset, which we do not have a control of.  

Page 5, section 3.1 (Dataset) is too vague. How many total possessions were in the dataset? How many games were these possessions from? From which NBA seasons were these games derived? All of these questions need to be answered in this section so that readers have a better understanding of the sample/size.

we have included more information about the dataset in the section 3.1 (lines 331 to 339, and lines 368 to 380) and a new subsection 3.3 to discuss reliability measures. 

The Discussion of Results section (4.2) is too short. Despite setting the study up to suggest that this application would be able to “explain the strategies employed by the defensive team,” I see very little discussion of basketball strategy stemming from these results. If this method could “be used to inform coaching decisions,” we need to see some examples of it in action. This last point is crucial for me. Tell me how this method—how having a machine learning method that can classify defensive “switch” and “trap” strategies—can make a big difference on the basketball scene.

Yes, we acknowledge this discussions section was too small, now we have included two completely new subsections 4.3 to discuss the application towards basketball scene and 4.4 to discuss the limitations of the current method.

To the previous point, are you simply attempting to classify defensive play as a “trap” or “switch” strategy? Or are you also classifying “man” and “zone” strategies? You need to be more clear about what exact strategies you are classifying. Furthermore, as mentioned in the second and third points above, there are a variety of hybrid/combo schemes beyond these four that at least merit a mention, even if merely as a limitation to your method.

We are classifying only switch and trap  strategies used against pick and roll. We appreciate this was not clear in the original text, but we have adjusted, the abstract, introduction, conclusions and section 3.2 to make this clearer. 

Furthermore, as suggested we have included the limitations of this study in section 4.4.

You also need to be more clear about your classifier (input) variables. Page 7, line 269, for example, says you considered “eleven features that contain the movement vectors of four players whose defensive relationship changes…” What are these eleven features? Most studies have a table listing the input variables with definitions, and this study would certainly benefit from such a table.

We have now included figure 6 (a) and (b) to illustrate the different features used in the machine learning models. Page 7, line 478 to 481.

The rationale/statistics for why you ultimately chose the SVM over KNN needs to be elaborated on more clearly. With everything you’ve presented, KNN looked like the more accurate method; give more evidence for SVM being the preferred solution.

A justification is given in page 13, lines 529 to 533.

"In the present experiments, the KNN shows the highest accuracy levels. However, it should be noted KNN algorithm works by comparing a new data point against all the past data points, and assigning it for the majority voted class within the k-number of nearest neighbors. Therefore, for the KNN algorithm to be used, all the data points need to be stored on the computer. This is a significant disadvantage, especially when the dataset is extremely large. .  Therefore, considering both the computational complexity and the classifier performance, the SVM with the movement vectors and distance features, would be the best method to classify defensive strategies"

Minor Suggestions:

There are some grammatical and typographical errors interspersed throughout the manuscript; a fine-toothed edit would be helpful before resubmission.

Yes, we have further read the paper to identify several grammatical and typographical errors.

The top-middle box in Figure 4 and the top-right box in Figure 6 have typographical errors.

Thanks for this comment, we have corrected these figures. 

On page 5, lines 189-90, the term “efficiency” should be used when describing performance on a per-possession basis, and further notation should be made as to why efficiency is important in basketball.

We have now included this term within the text (Page 7/ line 331 in the new manuscript). 

Make sure you are not using the terms “play” and “possession” interchangeably (e.g., Page 7, lines 255-256). These are two different things. Multiple plays can take place within a single possession.

We have identified several places in which this term was used interchangeably, and corrected this mistake. 

Additional Helpful References

Franks, A., Miller, A., Bornn, L., & Goldsberry, K. (2015, February). Counterpoints: Advanced defensive metrics for NBA basketball. In 9th Annual MIT Sloan Sports Analytics Conference, Boston, MA.

Kubatko, J., Oliver, D., Pelton, K., & Rosenbaum, D. T. (2007). A starting point for analyzing basketball statistics. Journal of Quantitative Analysis in Sports3(3).

Shea, S. M. (2014). Basketball analytics: Spatial tracking.

Thanks for the signposting, we have used all these references, and cited the papers appropriately at different parts of the paper, especially in section 2.2 and 4.3. 

Reviewer 2 Report

The main limitations of the manuscript are the poor scientific rationale and use of key references to justify their introduction (first and second paragraph) and discussion sections and the invalid and non-scientific use of basketball strategies. Then, I strongly suggest to use some of the following references focused on basketball and to redefine the basketball strategies:

Bourbousson, J., Sève, C., & McGarry, T. (2010). Space–time coordination dynamics in basketball: Part 1. Intra-and inter-couplings among player dyads. Journal of sports sciences, 28(3), 339-347. Bourbousson, J., Sève, C., & McGarry, T. (2010). Space–time coordination dynamics in basketball: Part 2. The interaction between the two teams. Journal of Sports Sciences, 28(3), 349-358. Courel-Ibáñez, J., McRobert, A. P., Toro, E. O., & Vélez, D. C. (2017). Collective behaviour in basketball: a systematic review. International Journal of Performance Analysis in Sport, 17(1-2), 44-64. Courel-Ibáñez, J., McRobert, A. P., Ortega Toro, E., & Cárdenas Vélez, D. (2018). Inside Game Effectiveness in Nba Basketball: Analysis of Collective Interactions. Kinesiology: International journal of fundamental and applied kinesiology, 50(2), 0-0. Esteves, P. T., Silva, P., Vilar, L., Travassos, B., Duarte, R., Arede, J., & Sampaio, J. (2016). Space occupation near the basket shapes collective behaviours in youth basketball. Journal of sports sciences, 34(16), 1557-1563. Gómez, M. A., Lorenzo, A., Ibañez, S. J., & Sampaio, J. (2013). Ball possession effectiveness in men's and women's elite basketball according to situational variables in different game periods. Journal of sports sciences, 31(14), 1578-1587. Gómez, M. Á., Battaglia, O., Lorenzo, A., Lorenzo, J., Jiménez, S., & Sampaio, J. (2015). Effectiveness during ball screens in elite basketball games. Journal of Sports Sciences, 33(17), 1844-1852. Lamas, L., Junior, D. D. R., Santana, F., Rostaiser, E., Negretti, L., & Ugrinowitsch, C. (2011). Space creation dynamics in basketball offence: validation and evaluation of elite teams. International Journal of Performance Analysis in Sport, 11(1), 71-84. Lamas, L., Santana, F., Heiner, M., Ugrinowitsch, C., & Fellingham, G. (2015). Modeling the offensive-defensive interaction and resulting outcomes in basketball. PloS one, 10(12), e0144435. Lorenzo Calvo, J., Menéndez García, A., & Navandar, A. (2017). Analysis of mismatch after ball screens in Spanish professional basketball. International Journal of Performance Analysis in Sport, 17(4), 555-562. Leite, N. M., Leser, R., Gonçalves, B., Calleja-Gonzalez, J., Baca, A., & Sampaio, J. (2014). Effect of defensive pressure on movement behaviour during an under-18 basketball game. International journal of sports medicine, 35(09), 743-748. Marmarinos, C., Apostolidis, N., Kostopoulos, N., & Apostolidis, A. (2016). Efficacy of the “pick and roll” offense in top level European basketball teams. Journal of human kinetics, 51(1), 121-129. Metulini, R., Manisera, M., & Zuccolotto, P. (2017). Space-Time Analysis of Movements in Basketball using Sensor Data. arXiv preprint arXiv:1707.00883. Metulini, R., Manisera, M., & Zuccolotto, P. (2018). Modelling the dynamic pattern of surface area in basketball and its effects on team performance. Journal of Quantitative Analysis in Sports, 14(3), 117-130. Ramos-Villagrasa, P. J., Navarro, J., & García-Izquierdo, A. L. (2012). Chaotic dynamics and team effectiveness: Evidence from professional basketball. European Journal of Work and Organizational Psychology, 21(5), 778-802. Sampaio, J., Leser, R., Baca, A., Calleja-Gonzalez, J., Coutinho, D., Gonçalves, B., & Leite, N. (2016). Defensive pressure affects basketball technical actions but not the time-motion variables. Journal of sport and health science, 5(3), 375-380. Sampaio, J., McGarry, T., Calleja-González, J., Sáiz, S. J., i del Alcázar, X. S., & Balciunas, M. (2015). Exploring game performance in the National Basketball Association using player tracking data. PloS one, 10(7), e0132894. Santana, F. L., Rostaiser, E., Sherzer, E., Ugrinowitsch, C., Barrera, J., & Lamas, L. (2015). Space protection dynamics in basketball: validation and application to the evaluation of offense-defense patterns. Motriz: Revista de Educação Física, 21(1), 34-44. Santana, F., Fellingham, G., Rangel, W., Ugrinowitsch, C., & Lamas, L. (2019). Assessing basketball offensive structure: The role of concatenations in space creation dynamics. International Journal of Sports Science & Coaching, 14(2), 179-189. Vaquera, A., García-Tormo, J. V., Gómez Ruano, M. A., & Morante, J. C. (2016). An exploration of ball screen effectiveness on elite basketball teams. International Journal of Performance Analysis in Sport, 16(2), 475-485.

The authors should rewrite the introduction trying to support the analysis of space and defensive actions in basketball. And then, the use of machine learning and sport analytics to analyse the data derived from tracking systems. The problem here is that the main aim is to use a “tool” (statistical approach) to solve a problem from practice (basketball behaviours), therefore the rationale should cover both issues.

The basketball strategies are not based on a scientific approach. Please justify using the previous/suggested references the different tactics and strategies. This reviewer considers that the authors did not select the correct contexts of analysis (some of them were omitted and only 3 contexts analysed). What about pressure defence or mixed defences? Pick and pop? Defensive flashes? The produce used is subjective and empirically invalid.

The dataset have to include the data reliability as a previous step to use the data.

The analytical model is incomplete, please read and use the references suggested in order to improve the rationale and procedure of this article.

Discussion, the article only covers the machine learning model but nothing about the key of the analysis: to identify and recognize defensive actions. Please, improve the section justifying the application of the main results and then the models used to classify the defences.

Author Response

Dear Reviewer,

Due to your comments, we have made several significant changes, in to the resubmission of the manuscript. We would like to thank your for your considerate and thoughtful comments which you have provided voluntarily. Your effort has enabled this paper to be massively improved. 

We have provided additional information necessary for understanding our method, and here we would like to point the key changes in the current manuscript:

We have included a new section (subsection 2.2) on the work carried out in basketball analytics in general, as it was important to position our contribution. A total of 24 new references are included in the new manuscript. The introduction is improved to provide more background about basketball analytics, previous work, its importance and how our work fit in.  More background on defensive mechanisms is provided in section 2.1. More information about the data set is provided in section 3.1, and the data reliability measures are included in the new subsection 3.3.  A new subsection (section 4.3) under discussions is written about the usage of this model and the contribution in general towards the basketball analytics area.  A new subsection 4.4 is included to write about the limitations of the current study.  The abstract and conclusion is adjusted to reflect the above changes and to narrow down the focus of defensive strategies in general to defensive strategies employed against pick and roll. 

We have left all the new additions with track changes to highlight the changes. 

Below you can find a point by point response to individual questions/comments posed by you. 

Thanks again,

Authors

Point by point responses

Review 2:

The main limitations of the manuscript are the poor scientific rationale and use of key references to justify their introduction (first and second paragraph) and discussion sections and the invalid and non-scientific use of basketball strategies. Then, I strongly suggest to use some of the following references focused on basketball and to redefine the basketball strategies:

We have addressed all the comments of the reviewers and have correctly focused our contribution. We acknowledge that the initial manuscript was not clear about the very specific contribution of the paper, which we have addressed in the current draft. We have now focused only on two very specific defensive strategies against pick and roll, and not suggesting to address all of defense in basketball. We have now changed the introduction to include most of the references suggested by you, and more references. Furthermore we have included all the references you have suggested in a new section under the background section 2. We sincerely hope this manuscript is a much improved version with a clear scientific rationale. 

Bourbousson, J., Sève, C., & McGarry, T. (2010). Space–time coordination dynamics in basketball: Part 1. Intra-and inter-couplings among player dyads. Journal of sports sciences28(3), 339-347. Bourbousson, J., Sève, C., & McGarry, T. (2010). Space–time coordination dynamics in basketball: Part 2. The interaction between the two teams. Journal of Sports Sciences28(3), 349-358. Courel-Ibáñez, J., McRobert, A. P., Toro, E. O., & Vélez, D. C. (2017). Collective behaviour in basketball: a systematic review. International Journal of Performance Analysis in Sport17(1-2), 44-64. Courel-Ibáñez, J., McRobert, A. P., Ortega Toro, E., & Cárdenas Vélez, D. (2018). Inside Game Effectiveness in Nba Basketball: Analysis of Collective Interactions. Kinesiology: International journal of fundamental and applied kinesiology50(2), 0-0. Esteves, P. T., Silva, P., Vilar, L., Travassos, B., Duarte, R., Arede, J., & Sampaio, J. (2016). Space occupation near the basket shapes collective behaviours in youth basketball. Journal of sports sciences34(16), 1557-1563. Gómez, M. A., Lorenzo, A., Ibañez, S. J., & Sampaio, J. (2013). Ball possession effectiveness in men's and women's elite basketball according to situational variables in different game periods. Journal of sports sciences31(14), 1578-1587. Gómez, M. Á., Battaglia, O., Lorenzo, A., Lorenzo, J., Jiménez, S., & Sampaio, J. (2015). Effectiveness during ball screens in elite basketball games. Journal of Sports Sciences33(17), 1844-1852. Lamas, L., Junior, D. D. R., Santana, F., Rostaiser, E., Negretti, L., & Ugrinowitsch, C. (2011). Space creation dynamics in basketball offence: validation and evaluation of elite teams. International Journal of Performance Analysis in Sport11(1), 71-84. Lamas, L., Santana, F., Heiner, M., Ugrinowitsch, C., & Fellingham, G. (2015). Modeling the offensive-defensive interaction and resulting outcomes in basketball. PloS one10(12), e0144435. Lorenzo Calvo, J., Menéndez García, A., & Navandar, A. (2017). Analysis of mismatch after ball screens in Spanish professional basketball. International Journal of Performance Analysis in Sport17(4), 555-562. Leite, N. M., Leser, R., Gonçalves, B., Calleja-Gonzalez, J., Baca, A., & Sampaio, J. (2014). Effect of defensive pressure on movement behaviour during an under-18 basketball game. International journal of sports medicine35(09), 743-748. Marmarinos, C., Apostolidis, N., Kostopoulos, N., & Apostolidis, A. (2016). Efficacy of the “pick and roll” offense in top level European basketball teams. Journal of human kinetics51(1), 121-129. Metulini, R., Manisera, M., & Zuccolotto, P. (2017). Space-Time Analysis of Movements in Basketball using Sensor Data. arXiv preprint arXiv:1707.00883. Metulini, R., Manisera, M., & Zuccolotto, P. (2018). Modelling the dynamic pattern of surface area in basketball and its effects on team performance. Journal of Quantitative Analysis in Sports14(3), 117-130. Ramos-Villagrasa, P. J., Navarro, J., & García-Izquierdo, A. L. (2012). Chaotic dynamics and team effectiveness: Evidence from professional basketball. European Journal of Work and Organizational Psychology21(5), 778-802. Sampaio, J., Leser, R., Baca, A., Calleja-Gonzalez, J., Coutinho, D., Gonçalves, B., & Leite, N. (2016). Defensive pressure affects basketball technical actions but not the time-motion variables. Journal of sport and health science5(3), 375-380. Sampaio, J., McGarry, T., Calleja-González, J., Sáiz, S. J., i del Alcázar, X. S., & Balciunas, M. (2015). Exploring game performance in the National Basketball Association using player tracking data. PloS one10(7), e0132894. Santana, F. L., Rostaiser, E., Sherzer, E., Ugrinowitsch, C., Barrera, J., & Lamas, L. (2015). Space protection dynamics in basketball: validation and application to the evaluation of offense-defense patterns. Motriz: Revista de Educação Física21(1), 34-44. Santana, F., Fellingham, G., Rangel, W., Ugrinowitsch, C., & Lamas, L. (2019). Assessing basketball offensive structure: The role of concatenations in space creation dynamics. International Journal of Sports Science & Coaching14(2), 179-189. Vaquera, A., García-Tormo, J. V., Gómez Ruano, M. A., & Morante, J. C. (2016). An exploration of ball screen effectiveness on elite basketball teams. International Journal of Performance Analysis in Sport16(2), 475-485.

Thanks for these suggestions, we have included a new section 2.2 under the heading of Data Driven Performance Analysis in Basketball to discuss the above references. The new section 2.2 captures the essence of the key performance analysis literature mentioned above. 

The authors should rewrite the introduction trying to support the analysis of space and defensive actions in basketball. And then, the use of machine learning and sport analytics to analyse the data derived from tracking systems. The problem here is that the main aim is to use a “tool” (statistical approach) to solve a problem from practice (basketball behaviours), therefore the rationale should cover both issues.

We have now changed the introduction to include most of the references suggested by you, and more references. Altogether there are 24 new references in the current text. 

We have made sure the paper is positioned within the current literature, as a method to recognize a given defensive play type from player and ball tracking, provided that training data can be procured.  

Furthermore, we have included a new subsection 4.3 to discuss the application of the methodology for basketball analytics. 

The basketball strategies are not based on a scientific approach. Please justify using the previous/suggested references the different tactics and strategies. This reviewer considers that the authors did not select the correct contexts of analysis (some of them were omitted and only 3 contexts analysed). What about pressure defence or mixed defences? Pick and pop? Defensive flashes? The produce used is subjective and empirically invalid.

We believe this issue was mainly raised because in our original draft we were not very explicit as to what defensive strategies we are addressing. This has been rectified in the current draft, and we have discussed this in section 2.1.1, and the next text can be found from lines 151 to 159

We are classifying only switch and trap  strategies used against pick and roll. We appreciate this was not clear in the original text, but we have adjusted, the abstract, introduction, conclusions and section 3.2 to make this clearer. We have discussed about pick-and-pop in lines 176 to 179, as a different offensive strategy. 

We have also discussed that pressure defense cannot be analyzed since we are only considering half court possession data, and now acknowledged as a limitation in lines 366 to 367. However, this is a limitation of the publicly available dataset, which we do not have a control of.  

The dataset have to include the data reliability as a previous step to use the data.

we have included more information about the dataset in the section 3.1 (lines 331 to 339, and lines 368 to 380) and a new subsection 3.3 to discuss reliability measures.

The analytical model is incomplete, please read and use the references suggested in order to improve the rationale and procedure of this article.

We have improved the presentation of the model, with new information about the dataset preparation in lines 368 to 377,  and clearly informed the need for the analytical model in the new lines 382 to 384

Most importantly, we have positioned current contribution within the grand scheme of basketball analytics more appropriately. 

Discussion, the article only covers the machine learning model but nothing about the key of the analysis: to identify and recognize defensive actions. Please, improve the section justifying the application of the main results and then the models used to classify the defences.

Yes, we acknowledge this discussions section was too small, now we have included two completely new subsections 4.3 to discuss the application towards basketball scene and 4.4 to discuss the limitations of the current method.

Round 2

Reviewer 2 Report

Some minor issues should be addressed in order to improve the quality of this study:

L140 and L141, please write triangle and two (5 players defending).

L300, L303, L304, please write 36,330.

L304, please write 32,377.

L326, please write 42,865

L327, please write 32,377.

L394, please write 32,378.

L394 please clarify why that sample is different (1 unit) to the previous one explained in methods section.

L552, please write 32,377.

Two references should be added to justify the sport analytic section:

Metulini, R., Manisera, M., & Zuccolotto, P. (2018). Modelling the dynamic pattern of surface area in basketball and its effects on team performance. Journal of Quantitative Analysis in Sports, 14(3), 117-130.

Sampaio, J., Gonçalves, B., Mateus, N., Shaoliang, Z., & Leite, N. (2018). 6 Basketball. Modelling and simulation in sport and exercise.

Author Response

Dear Reviewer,

We have addressed these minor issues as suggested. 

We have found the line numbers to be inconsistent at some points, and thus made the best guess to identify the correct line number. 

Thank you for your effort. 

Yours,

Authors.

Reviewer 2: Comments (Response in orange)

Some minor issues should be addressed in order to improve the quality of this study:

L140 and L141, please write triangle and two (5 players defending).

Done

L300, L303, L304, please write 36,330.

L304, please write 32,377.

L326, please write 42,865

L327, please write 32,377.

Done

L394, please write 32,378.

We have changed everything to be 32,377. 

L394 please clarify why that sample is different (1 unit) to the previous one explained in methods section.

This was a mistake to have it as 32,378. We have corrected it to 32377.

L552, please write 32,377.

Yes, this was included in the corrections. as (e.g. 32,377) 

Two references should be added to justify the sport analytic section:

Metulini, R., Manisera, M., & Zuccolotto, P. (2018). Modelling the dynamic pattern of surface area in basketball and its effects on team performance. Journal of Quantitative Analysis in Sports14(3), 117-130.

Included: Line 246 in the current draft

Sampaio, J., Gonçalves, B., Mateus, N., Shaoliang, Z., & Leite, N. (2018). 6 Basketball. Modelling and simulation in sport and exercise.

Included: Lines 203 to 205 in current draft